# Corruption in Organizations: Ethical Climate and Individual Motives

**Madelijne Gorsira** [1,*] ![ID], **Linda Steg** [2], **Adriaan Denkers** [1] and **Wim Huisman** [1]

1  Department of Criminal Law and Criminology, VU University Amsterdam, 1081 HV Amsterdam,
   The Netherlands; adriaandenkers@icloud.com (A.D.); w.huisman@vu.nl (W.H.)
2  Department of Psychology, University of Groningen, 9712 TS Groningen, The Netherlands; e.m.steg@rug.nl
*  Correspondence: m.gorsira@vu.nl

**Abstract:** The aim of this research was to examine how organizational and individual factors, in concert, shape corruption. We examined whether the ethical climate of organizations is related to corruption, and if so, whether it affects corruption *through* individual motives for corruption. A large-scale questionnaire study was conducted among public officials ($n$ = 234) and business employees ($n$ = 289) who were in a position to make corrupt decisions. The findings suggest that public and private sector employees who perceive their organizational climate as more egoistic and less ethical are more prone to corruption. This relationship was fully mediated by individual motives, specifically by personal and social norms on corruption. These results indicate that employees who perceive their organization's ethical climate as more egoistic and less ethical experience weaker personal and social norms to refrain from corruption, making them more corruption-prone. Hence, strategies addressing the interplay between organizational factors and individual motives seem promising in curbing corruption. To effectively withhold employees from engaging in corruption, organizations could deploy measures that strengthen an organizations' ethical climate and encourage ethical decision-making based on concern for the wellbeing of others, as well as measures increasing the strength of personal and social norms to refrain from corruption.

**Keywords:** bribery; corruption; ethical climate; organizations; personal and social norms

## 1. Introduction

Increasingly, organizations are being held responsible for their employees' unethical and illegal behavior (Victor and Cullen 1988; Wells 2014). This is especially true for corruption. For example, under the United Kingdom Bribery Act 2010, UK-connected companies can be held criminally liable for employees' acts of bribery unless the company can prove it had in place 'adequate procedures'[1] designed to prevent bribery (Lord and Levi 2016). In February 2016, the company Sweett Group PLC was the first to be convicted[2] of having failed to take such adequate steps, and was ordered to pay £2.25 million.[3] As a result, organizations are becoming progressively concerned about[4] and are taking steps[5] to prevent bribery in their organization. Yet, which measures are actually effective in preventing

---

1  www.legislation.gov.uk/ukpga/2010/23/section/7.
2  This was the first conviction, but not the first case to be dealt with using the s.7 corporate offence. In November 2015, the Serious Fraud Office (SFO) and Standard Bank reached a deferred prosecution agreement (DPA) for a failure to prevent bribery, contrary to section 7 of the Bribery Act 2010 (www.sfo.gov.uk/2015/11/30/sfo-agrees-first-uk-dpa-with-standard-bank/).
3  www.sfo.gov.uk/2016/02/19/sweett-group-plc-sentenced-and-ordered-to-pay-2-3-million-after-bribery-act-conviction/.
4  www.willis.com/documents/publications/Industries/Financial_Institutions/Directors_Liability.pdf.
5  www.ey.com/Publication/vwLUAssets/EY-managing-fraud-bribery-and-corruption-risks-in-the-mining-and-metals-industry/$FILE/EY-managing-fraud-bribery-and-corruption-risks-in-the-mining-and-metals-industry.pdf.

corruption is largely unknown. Hence, a key question is which factors cause corruption, and which strategies may be effective in preventing it.

The vast majority of empirical corruption studies focus on differences between countries (Dong et al. 2012; Svensson 2003), and look at macro-level determinants of corruption, such as political institutions (Lederman et al. 2005), and the culture in a country (Jha and Panda 2017). Such research may offer explanations for why corruption is more common in some countries than in others. It does not, however, reveal why within countries corruption seems to flourish more in some sectors and organizations than in others, nor why within organizations some people engage in corruption while their colleagues do not. Furthermore, country-level factors are generally very stable and difficult to alter. As a result, explanations on the macro level may not be very helpful in designing practical anti-corruption measures (Rose-Ackerman 2010). Hence, a better understanding of why certain individuals within certain organizations commit corruption seems vital. For this reason, we examine which organizational and individual factors may explain employees' engagement in corruption. Hence, we aim to identify the 'bad barrels' and the 'bad apples' (Kish-Gephart et al. 2010). More specifically, we conduct a questionnaire study among a large sample of employees who are likely to face corruption-prone situations, as questionnaires can provide insight into multiple key correlates of corruption (Andvig et al. 2001). Corruption is most often defined as the abuse of public power for private benefit (Aguilera and Vadera 2008; Tanzi 1998). This definition includes many unethical behaviors, such as embezzlement, conflicts of interests, and forgery. In this paper, we focus on a specific form of corrupt behavior: bribery. Bribery is not only unethical but also illegal; it is criminalized in both national and international legislation (e.g., The United Kingdom Bribery Act 2010 and the United Nations Convention Against Corruption). The present study takes both sides of bribery, the giver and the taker, into account. Bribery most commonly occurs in interactions between public and private sectors (Rose-Ackerman 1997; Rose-Ackerman 2007). When bribery involves a public official and a business employee, it is likely that the latter bribes the former. If so, the business employee engages in *active* bribery and the public official in *passive* bribery (Beets 2005; Huberts and Nelen 2005). In this paper, we therefore focus on the bribery of public officials by business employees. Below, we first review the empirical literature concerning individual factors influencing corruption. Next, we discuss a prominent organizational factor that may affect corruption. Subsequently, we discuss how these individual and organizational factors might be related, and how they, in concert, may shape corruption.

## 1.1. Individual Factors Explaining Corruption

Corruption ultimately results from decisions made by individuals. Various disciplines have postulated explanations for why some individuals are more prone to corruption than others, specifically economics, criminology, and social psychology (Gorsira et al. 2016). The most prominently explanations include incentives (Dong et al. 2012; Prabowo 2014; Shover and Bryant 1993; Andvig et al. 2001; Dimant 2013), opportunities (see, for instance, Aguilera and Vadera 2008; Pinto et al. 2008; Graycar and Sidebottom 2012), and norms (see, for instance, Powpaka 2002; Köbis et al. 2015; Rabl and Kühlmann 2008; Tavits 2010). A recent study demonstrated that all of these factors were, indeed, related to proneness to corruption (Gorsira et al. 2016). The study showed that public officials and business employees who perceived higher benefits of corruption, e.g., financial gains, excitement, and pleasure, and perceived lower costs of corruption, i.e., a lower chance of detection, and a less severe punishment, were more prone to engage in corruption (Gorsira et al. 2016). Similarly, employees who perceived more opportunities to engage in, and less opportunities to refrain from, corruption reported to be more prone to it. Employees who reported weaker personal norms, i.e., who felt less morally obliged to refrain from corruption, and who reported weaker social norms, i.e., those who thought their close colleagues approved of and engaged in corruption, also reported to be more prone to corruption. The outcomes of this study further suggest that the motives that contribute uniquely to employees' proneness to corruption, when other motives were controlled for, were the perceived opportunity to refrain from corruption and personal and social norms on corruption. Importantly,

the pattern of results was identical for public and private sector respondents when business employees were asked about their proneness to active bribery, and public officials about their proneness to passive bribery. This indicates that the same motives underlie both sides of bribery (Gorsira et al. 2016). In the present study, we expect these individual motives to be related to public officials' and business employees' proneness to corruption.

### 1.2. Organizational Factors Explaining Corruption

Individuals do not operate in a vacuum; corruption occurs within an organizational context. Hence, it seems important to not only look at the 'bad apples', but also at possible 'bad barrels'. A relevant organizational-level predictor of (un)ethical behavior in organizations is the culture (Jha and Panda 2017; Kaptein 2011)[6] or climate of an organization (Victor and Cullen 1988). One meta-analysis revealed an organizations' ethical climate as a particularly relevant organizational factor explaining a wide range of unethical decisions of employees (Kish-Gephart et al. 2010; Peterson 2002).[7] We, therefore, focus on the ethical climate, rather than on ethical culture. An organization's ethical climate affects which issues organization members consider ethically relevant, whose interests they consider when deciding on moral issues, and which ethical criteria they use to determine what constitutes the 'right behavior' (Victor and Cullen 1988; Martin and Cullen 2006). As such, ethical climate refers to the most commonly used *types* of moral reasoning, rather than to the content of decisions (Victor and Cullen 1988). The assumption underlying the concept of ethical climate is that members of an organization share, at least to some extent, a form of moral reasoning.

In deciding whether it is acceptable to pay or to accept bribes, organizational members can consider different ethical criteria. The dominant considerations are maximizing self-interest (egoistic reasoning), maximizing joint interests (benevolent reasoning), or adherence to principles (principled reasoning; Victor and Cullen 1988). In an egoistic climate, people base their decisions, first and foremost, on what will best promote their self-interest (Martin and Cullen 2006). In such a climate, organizational members perceive that self-interest commonly guides behavior, even if it is at the expense of others (Wimbush and Shepard 1994). In a benevolent climate, ethical decision-making is seen to be predominantly based on concern for the wellbeing of others, which may be others within the organization itself, or society at large (Martin and Cullen 2006). In a principled climate, ethical decision-making is primarily presumed to be based on external codes, such as the law or professional codes of conduct (Martin and Cullen 2006).

More recently, a simplified model and measure of ethical climate has been postulated (Arnaud 2010; Arnaud and Schminke 2006, 2012; Abernethy et al. 2012). This model proposes that organizational members typically base ethical decisions either on what best promotes their self-interest, or the interests of others. If the former form of moral reasoning is seen to predominate within an organization or department, the ethical climate can be characterized as primarily self-focused, which is likely to inhibit ethical behavior. If the latter is perceived to prevail, the ethical climate within the organization or department is predominantly other-focused, stimulating ethical behavior (Arnaud 2010; Arnaud and Schminke 2006). In the current paper, we use the terms 'egoistic climate' to refer to the former and 'ethical climate' to refer to the latter. Notably, we regard the two organizational-climate types as two ends of a single dimension, as it seems unlikely that an organizational climate is predominantly focused on maximizing self-interest and, at the same time, on maximizing the interests of others.

Empirical studies have generally found that employees who perceive their organizational climate as more egoistic report more unethical behavior, while employees who perceive their organizational

---

[6] Ethical culture is defined as "the perception about the conditions that are in place in the organization to comply or not comply with what constitutes unethical and ethical behavior", e.g., role-modeling of managers, and rewards and punishment (Kaptein 2011, p. 846).

[7] When ethical climate and ethical culture are considered as predictors simultaneously, ethical culture does not contribute uniquely to (un)ethical behavior (Kish-Gephart et al. 2010).

climate as more ethical report less unethical behavior (Kish-Gephart et al. 2010; Peterson 2002; Wimbush and Shepard 1994; Treviño et al. 1998; Mayer 2014).

Many scholars have also proposed a link between an organization's ethical climate and, specifically, corruption (Pinto et al. 2008; Grieger 2006; Motwani et al. 1998; Simha and Cullen 2012; Hess 2015; Martin et al. 2009). Few, however, have actually demonstrated one. One study found support for a relationship between an egoistic climate and corruption in organizations (Stachowicz-Stanusch and Simha 2013). Another study found that employees who perceived their organizational climate as more ethical were less likely to give gifts or favors in exchange for preferential treatment, whereas the opposite was true for a more egoistic climate (Peterson 2002). We examine whether perceived ethical climate is related to engagement in corruption, in particular bribery, among both public officials and business employees, thereby focusing on both sides of bribery. We propose the following hypothesis:

**Hypothesis 1 (H1).** *The more ethical and the less egoistic public and private sector employees perceive their organizational climate to be, the less likely they are to engage in corruption.*

### 1.3. Interplay

An important question is whether and how ethical climate is related to individual corruption motives. We propose that corruption is the result of an *interplay* between individual and organizational factors. Notably, individual motives for corruption and ethical climate are likely to be related, and may, in concert, shape corruption. To date, studies on individual and organizational explanations for corruption have almost exclusively focused on either individual or organizational factors, and have examined the relationship of each factor to corruption independently (Den Nieuwenboer and Kaptein 2008). Studies that simultaneously include organizational and individual factors will enhance our understanding of why employees engage in corruption. For the first time, we aim to systematically investigate how individual and organizational factors are related not only to corruption, but also to each other. In particular, we examine how perceived ethical climate, which is typically considered to be an *organizational* characteristic (Arnaud 2010; Arnaud and Schminke 2006, 2012), is related to *individual* motives for *corruption* as a specific type of unethical behavior. We define ethical climate as one's perceptions of how people within an organization typically make ethical decisions in general. As such, ethical climate is likely to affect many ethical and unethical behaviors, including corruption, which is why we propose that the perceived organizational ethical climate may affect individual motives for corruption as a specific form of unethical behavior. Therefore, the perceived general ethical climate of organizations is likely to affect motives for various types of unethical behavior, including corruption of employees, rather than the other way around. More specifically, we hypothesize that perceived ethical climate is related to corruption *via* individual motives for corruption. In particular, ethical climate may weaken or strengthen the motives for corruption, thereby, increasing or decreasing the likelihood of corruption.

We propose that ethical climate particularly influences personal and social norms on corruption. Notably, ethical climate refers to the *most commonly* used types of *moral* reasoning within organizations (Victor and Cullen 1988; Arnaud 2010). As such, ethical climate may shape employees' personal and social norms on corruption. Some scholars indeed have speculated that ethical climate affects (un)ethical behavior through individual factors, particularly ethics-related ones (Kish-Gephart et al. 2010; Treviño et al. 1998; Webb 2012). Specifically, it has been proposed that awareness of moral obligation functions as a mediator and, notably, that an organization's ethical climate may strengthen or weaken employees' personal norms, which in turn affect the likelihood of unethical behavior (Wang and Hsieh 2013; Guthrie et al. 2006). Furthermore, there is initial empirical evidence to suggest that ethical climate affects unethical behavior (i.e., employees' illegal copying of software) through social norms on the behavior (Lin et al. 1999). To date, however, little is known

about the mechanisms through which ethical climate exerts influence on the unethical and, more specifically, corrupt behavior of employees. In short, although it is unclear how ethical climate affects employees' unethical behavior and corrupt behavior specifically, ethical climate perceptions might especially influence normative motives, such as personal and social norms, which in turn, affect the likelihood of corruption. Hence, we aim to examine whether a more ethical and less egoistic climate elicits stronger personal and social norms to refrain from corruption, which, in turn, decreases the likelihood of corruption.

Additionally, we explore whether perceived ethical climate is related to other motives for corruption, in particular, the perceived costs and benefits of corruption and perceived opportunities to engage in and to refrain from corruption, and whether these motives also mediate the relationship between ethical climate and corruption.

On the basis of our reasoning above, we hypothesize that:

**Hypothesis 2 (H2).** *The more ethical and the less egoistic public and private sector employees perceive their organizational climate to be, in general, the stronger their personal and social norms to refrain from corruption, specifically, in turn resulting in less corruption.*

## 2. Materials and Methods

To gain a better insight into the key correlates of corruption, we conducted a large-scale study among a large sample of people who, in all likelihood, were in a position to bribe or to be bribed. That is, public officials and company employees who regularly interacted professionally with employees of the other sector, and who performed corruption-sensitive tasks. Our study was conducted in the Netherlands, which, according to the Corruption Perceptions Index of Transparency International, belongs to the least corrupt nations worldwide (Transprancy International 2016). In line with this, the number of Dutch people who report having had to pay a bribe is low compared to other European countries.[8] Nonetheless, the same survey indicates that more than half of the Dutch respondents think corruption is a widespread phenomenon in their country. Another study revealed that approximately 20%[9] of Dutch public officials reported having engaged in bribery-related behaviour in the past and/or to have an intention to do so in the near future (Gorsira et al. 2016).

### 2.1. Procedure and Respondents

A questionnaire study, preceded by a selection study, was conducted among members of a panel managed by an agency specializing in online research (www.flycatcher.eu[10]). Prospective participants had to meet the following criteria: they were employed either in the public or private sector; regularly interacted professionally with employees of the other sector (public officials with business employees and vice versa); and performed tasks over which they had discretionary powers.[11] In the selection study, 4318 panel members participated (a 70% response rate). On the basis of the selection criteria,

---

[8]   http://ec.europa.eu/commfrontoffice/publicopinion/archives/ebs/ebs_397_en.pdf.
[9]   Note that the respondents were selected, among others, on the basis of their discretionary powers.
[10]  The Flycatcher panel consists of approximately 16,000 members who have agreed to participate regularly in online surveys. On average, panel members receive eight surveys a year and, in exchange for completing the questionnaires, receive a small reward in the form of points, which can be converted into gift vouchers. The Flycatcher panel meets the ISO quality standards for social science research and is used exclusively for research, and not for any other purposes such as sales or direct marketing. Panel members may terminate their membership at any time and cannot select the type of surveys for which they wish to be invited.
[11]  A similar study was performed in 2013 by Gorsira et al. (2016). Panel members who participated in 2013 and who were still members of the Flycatcher-panel received an invitation to participate again. Of the 202 public officials who took part in 2013, 144 participated in the selection study (3.3%), of whom 73 participated in the main study (13.7%). Of the 200 business employees who took part in 2013, 140 participated in the selection study (3.2%), of whom 78 participated in the main study (14.9%). Hence, in total, 28.9% of the participants in the current study had participated in the 2013 study as well.

842 people received an invitation to participate in the questionnaire study.[12] Participation in the study was voluntary and anonymous. Given the sensitivity of the subject, the introduction stated that a study was being conducted by the Faculty of Law on integrity at work, rather than specifying that it was conducted by the Department of Penal Law and Criminology on corruption. The questions were presented in a randomized order, to counter order effects. To avert missing data, all questions had to be answered. A data quality check was performed on completion time, consistency of answers, and straight lining and, on the basis of this, 26 respondents were excluded due to poor response quality. The final sample, after the data quality check, consisted of 234 public officials and 289 business employees (a 62% response rate). Of the respondents to the questionnaire study, 53% were male. The participants' age ranged from 21 to 77 years, with a mean age of 44.8 (SD = 11.86). Compared to the general Dutch population, people with a higher level of education and income were overrepresented, which was expected, as the participants were selected on the basis of their discretionary powers, among other criteria.

Forty percent of respondents from the public sector interacted professionally with the private sector on a daily basis, 36% at least weekly, and 24% at least monthly. Of the respondents from the private sector, this was 37%, 54%, and 8%, respectively. These contacts were related to matters such as awarding contracts, purchasing goods and services, and enforcement and inspection, among others. On average, the public-sector respondents had been working at their organization for 4.3 years, at their current department for 3.5 years, and in their current function for 3.5 years. Of the private-sector respondents, this was 3.9 years, 3.6 years, and 3.5 years, respectively. Of the public-sector respondents, 22% held a management position and, of the private sector respondents, this was 31%.

*2.2. Measures*[13]

All measures were directed at the work context. The items measuring perceived ethical climate were derived from an instrument developed by Arnaud (2010). These items were the same for the public and private sectors. The items measuring motives for corruption and proneness to corruption were measured using a slightly modified version of a questionnaire developed by Gorsira et al. (2016).[14] This questionnaire consisted of two versions, one for the private sector and one for the public sector, the active and passive side of bribery respectively. Below, for each scale, we first provide an exemplary item for the private sector and then for the public sector. Unless otherwise noted, all items were scored on a 7-point scale ranging from strongly disagree (1) to strongly agree (7). Cronbach's alpha for all measures, as well as the means and standard deviations, are reported separately for the public and private sectors in Table 1.

**Table 1.** Summary statistics, disaggregated for the public sector (*n* = 234) and the private sector (*n* = 289).

| Variables | Public Sector | | | Private Sector | | |
|---|---|---|---|---|---|---|
| | $\alpha$ | M | SD | $\alpha$ | M | SD |
| Corruption-proneness | 0.90 | 0.19 | 0.392 | 0.93 | 0.21 | 0.409 |
| Perceived ethical climate | 0.89 | 5.05 | 1.071 | 0.88 | 4.92 | 1.062 |
| Personal norms on corruption | 0.78 | 5.64 | 0.969 | 0.85 | 5.56 | 1.126 |
| Social norms on corruption | 0.76 | 4.87 | 1.111 | 0.83 | 5.70 | 1.133 |
| Possibility to engage in corruption | 0.61 | 3.22 | 1.370 | 0.65 | 2.66 | 1.342 |
| Possibility to refrain from corruption | 0.69 | 5.86 | 0.973 | 0.63 | 5.54 | 1.081 |

---

[12]  The Ethics committee granted permission to conduct the study and, since fully disclosing the purpose of the study upfront could alter responses, waived the need to obtain participants' written consent.

[13]  We only elaborate on the measures that are relevant for the current study.

[14]  The full questionnaire is available from the first author upon request.

**Table 1.** *Cont.*

| | Public Sector | | | Private Sector | | |
|---|---|---|---|---|---|---|
| Variables | $\alpha$ | M | SD | $\alpha$ | M | SD |
| Costs of corruption | 0.90 | 4.70 | 1.118 | 0.90 | 4.72 | 1.233 |
| Benefits of corruption | 0.84 | 2.36 | 1.070 | 0.92 | 2.53 | 1.304 |
| Social desirability | 0.83 | 5.66 | 0.937 | 0.86 | 5.55 | 1.016 |

### 2.3. Dependent Variable

*Corruption* was operationalized by probing bribery-related intention and behavior, without using the words 'corruption' or 'bribery'. Three items measured bribery-related intentions ("In the foreseeable future, I can imagine that at my work a situation could arise in which I offer/give/promise money, goods or services to a public official in exchange for preferential treatment" (for the private sector); and "In the foreseeable future, I can imagine that at my work a situation could arise in which I ask/accept/expect money, goods or services in exchange for preferential treatment" (for the public sector)) on a scale ranging from 1 'not at all' to 7 'to a great extent'. Three similar items measured past bribery-related behavior ("At my work, I have offered/gave/promised money, goods or services to a public official in exchange for preferential treatment" (for the private sector); and "At my work, I have asked/accepted/expected money, goods or services in exchange for preferential treatment" (for the public sector)) on a scale ranging from 1 'never' to 7 'often'. The two scales were strongly correlated ($r = 0.71$, $p < 0.001$). Therefore, they were combined into one scale measuring corruption-proneness. As the six items formed a reliable scale, mean scores were computed. The average scores across the six items indicated that the respondents from both sectors reported themselves not to be very corruption-prone. As the data were not normally distributed, the scale was dichotomized to a corruption-prone category (consisting of respondents with a score of four or higher on the intention scale, and a score of two or higher on the past behavior scale[15]) and a non-corruption-prone category. Of the respondents, 19% of public and 21% of private sector respondents were categorized as corruption-prone, while the others were classified as non-corruption-prone.

### 2.4. Independent Variables

*Ethical climate* of respondents' own department was measured using a 10-item instrument (Arnaud 2010; Arnaud and Schminke 2006; Arnaud and Schminke 2012), with five items reflecting an egoistic climate (e.g., "In our department, people are mostly out for themselves" and "People around here protect their own interest above other considerations") and five items reflecting an ethical climate (e.g., "The most important concern is the good of all the people in the department" and "In our department, it is expected that you will always do what is right for society"). After recoding the items relating to egoistic climate, mean scores for the 10 items were computed, which formed an internally reliable scale, where the higher the score, the more the organizational climate was perceived as ethical rather than egoistic. As Table 1 indicates, according to both public and private sector respondents, the ethical climate of their respective organizations could be characterized as more ethical than egoistic.

*Personal norms on corruption* were measured by ten items (e.g., "I would feel guilty if I gave a public official money, goods or services in exchange for preferential treatment" (for the private sector); "I would feel guilty if I gave somebody from outside of my organization preferential treatment in exchange for money, goods or services" (for the public sector); and "I think it is over the top to have rules about accepting or offering gifts to public officials" (for both sectors). The 10 items formed

---

[15] We decided that respondents with a score of less than four on the intention scale could not conclusively be regarded as corruption-prone. With regard to self-reported corrupt behavior in the past, however, we reasoned that someone either had or had not engaged in bribery-related behavior; consequently, respondents with a score of two or higher on the behavior scale were classified as corruption-prone.

a reliable scale. Hence, mean scores were computed. The mean scores indicated that respondents from both sectors felt morally obliged to refrain from corruption; hence, on average, respondents experienced strong personal norms on corruption.

*Social norms on corruption* were measured by six items (e.g., "I am convinced that my close colleagues sometimes give money, goods or services to public officials in exchange for preferential treatment",[16] and "I am convinced that my close colleagues would feel guilty if they gave a public official money, goods or services in exchange for preferential treatment" (for the private sector); and "I am convinced that my close colleagues sometimes give somebody from outside our organization preferential treatment in exchange for money, goods or services",[17] and "I am convinced that my close colleagues would feel guilty if they gave somebody from outside our organization preferential treatment in exchange for money, goods or services" (for the public sector)). The six items formed a reliable scale. Therefore, mean scores were computed. The mean scores indicated that respondents of both sectors expected their close colleagues to disapprove of and refrain from corruption.

*Perceived opportunities to engage in corruption* were measured by three items (e.g., "There are many occasions during my work where I could bribe public officials" (for the private sector); "There are many occasions during my work where I could be bribed" (for the public sector); and "The rules on bribery at my work are easy to avoid" (for both sectors)). The three-item scale had a satisfactory reliability. Therefore, average scores were calculated, which indicated that, in both sectors, respondents did not perceive many opportunities to engage in corruption.

*Perceived opportunities to refrain from corruption* were measured by five items (e.g., "I am well aware of the rules about giving money, goods or services to public officials (for the private sector); "I am well aware of the rules about accepting money, goods or services of business contacts" (for the public sector); and "It is difficult to comply with bribery rules at my work"[18] (for both sectors)). The five-item scale had a satisfactory reliability. Therefore, mean scores were computed, which indicated that respondents perceived it as easy to refrain from corruption.

*Costs of corruption* were measured by 12 items, measuring the perceived chance of detection (six items; "Imagine that it is discovered that you engaged in bribery. In your opinion, how likely is it that the following persons or agencies would discover this . . . e.g., a direct colleague of yours; a supervisor from your organization; an enforcement agency (for both sectors)), and the severity of punishment (six items; "Imagine that it is discovered that you engaged in bribery. In your opinion, how serious would the negative consequences be, if the discovery was made by . . . e.g., a direct colleague of yours; a supervisor from your organization; an enforcement agency (for both sectors)). Responses were given on a 7-point scales ranging from 1 'not likely at all/not serious at all' to 7 'very likely/very serious'. The twelve items formed a reliable scale and the mean scores were computed, which indicated that respondents from both sectors assessed the costs of engaging in corruption as relatively high.

*Benefits of corruption* were measured by eleven items measuring how likely it is, in the respondents' perception, that someone would initiate, or go along with, a corrupt exchange (three items; e.g., "How likely do you think it is that you might get preferential treatment from a public official if you would offer him or her money, goods or services" (for the private sector); and "How likely do you think it is that someone from outside your organization would offer you money, goods or services to receive preferential treatment" (for the public sector)), and the benefits this would render (eight items; e.g., "Engaging in bribery would . . . lead to financial gain."; . . . make my job more exciting."; . . . lead to fun and pleasure." (for both sectors)). Responses were given on a 7-point scale ranging from 1 'very unlikely/strongly disagree' to 7 'very likely/ strongly agree'. The eleven items formed a reliable scale. Therefore, mean scores were computed. The mean scores indicated that respondents from both sectors perceived the benefits of engaging in corruption as not very high.

---

[16] This item was reversed scored during scale construction.
[17] This item was reversed scored during scale construction.
[18] This item was reversed scored during scale construction.

### 2.5. Control Variable

*Social desirability* was measured to control for respondents' tendencies to deny undesirable beliefs or behavior; a risk of special concern in ethics research (Fukukawa 2002). The Marlowe-Crowne Social Desirability Scale (Crowne and Marlowe 1960) has been widely used to test for the presence of this type of response; however, the items of this scale are rather general (e.g., "I sometimes think when people have a misfortune they only got what they deserved"). Since all items in the current study are directed at people's working situation, a social desirability scale was used that was specifically directed at a work context (Gorsira et al. 2016). Social desirability was measured by seven items (e.g., "At my work it has happened to me that I . . . benefitted from someone else"; " . . . took something (even a pen or a pin) that wasn't mine"; " . . . did not keep a promise" (for both sectors)). Responses were given on a 7-point scale ranging from 1 'never' to 7 'often'.[19] Mean scores were computed, which formed an internally reliable scale, and indicated that the respondents from both sectors responded in a rather socially desirable manner.

Independent-samples-*t*-tests were performed to investigate whether the two sectors differed regarding the average scores on the measures included in this study. The results suggested that public and private sector respondents did not significantly differ with regard to mean scores on: corruption-proneness; perceived ethical climate; personal norms; perceived costs of corruption; perceived benefits of corruption; and social desirability. However, compared to private sector respondents, public sector respondents perceived weaker social norms, which suggests that the public officials in the sample perceived corrupt behavior to be relatively more approved of and more common among their close colleagues compared to private sector respondents ($t(521) = 8.42$, $p < 0.001$). In addition, public sector respondents perceived more opportunities to engage in corruption ($t(521) = 4.66$, $p < 0.001$). Private sector respondents, on the other hand, perceived less opportunities to refrain from corruption than public sector respondents ($t(521) = 3.54$, $p < 0.001$).

### 2.6. Statistical Analyses

First, simple correlation coefficients were calculated for the public and private sectors, separately, to explore relationships between the variables included in the study. Next, a series of binary regression analyses were conducted over both sectors to test (a) whether perceived ethical climate explained corruption-proneness; (b) which motives for corruption uniquely explained corruption-proneness; and (c) whether the relationship between perceived ethical climate and corruption-proneness weakened or became statistically non-significant when the motives for corruption were included, the latter testing whether individual motives mediated the relationship between perceived ethical climate and corruption-proneness. Subsequently, for each motive, separately, we tested via bootstrapping (Zhao et al. 2010) which motives functioned as mediators in the relationship between perceived ethical climate and corruption-proneness. Additional analyses were performed to examine whether the same motives functioned as mediators in both the public and private sectors.

### 3. Results

The simple correlation coefficients between perceived ethical climate, individual motives for corruption, and corruption-proneness are displayed separately for the public and private sectors in Table 2. In both sectors, perceived ethical climate was negatively related to corruption. Hence, public and private sector respondents who perceived their organizational climate as more ethical and less egoistic reported to be less prone to corruption. This confirms the first hypothesis. In line with the results from another previous study (Gorsira et al. 2016), motives for corruption were significantly related to public and private sector respondents' corruption-proneness in the expected direction, except

---

[19]　All items were reversed scored during scale construction.

for perceived opportunities to engage in corruption in the public sector, which was not related to self-reported corruption-proneness. Furthermore, the results showed that perceived ethical climate was significantly related to personal and social norms on corruption; the more ethical and less egoistic public and private sector employees perceived their organizational climate to be, the more they felt morally obliged to refrain from corruption and the more they perceived their close colleagues to disapprove of and refrain from corruption. Furthermore, the results indicated that the more ethical and less egoistic public officials and business employees perceived their organizational climate to be, the less opportunities they perceived to engage in corruption; the more opportunities they perceived to refrain from corruption; the higher they assessed the costs of engaging in corruption; and the lower they assessed the benefits of corruption (but this relationship was only statistically significant for private sector respondents).[20] However, Table 2 shows that social desirability was significantly related to self-reported proneness to corruption in both sectors, as well as to perceived ethical climate in the private sector.[21] In the subsequent analyses, the influence of social desirability tendencies was, therefore, controlled for. Since the pattern of results appeared to be rather similar in both sectors, in order to enhance statistical power and to provide an overall view, the following analyses were performed over both groups and, thus, sectors, as the sector the respondents were employed in was included as a covariate.

**Table 2.** Simple correlations between corruption-proneness, ethical climate, and the motives for corruption, disaggregated for the public sector ($n = 234$) and the private sector ($n = 289$).

| | Corruption-Proneness | | Ethical Climate | |
|---|---|---|---|---|
| | **Public Sector** | **Private Sector** | **Public Sector** | **Private Sector** |
| Ethical climate | −0.18 ** | −0.26 *** | | |
| Personal norms on corruption | −0.30 *** | −0.41 *** | 0.34 *** | 0.46 *** |
| Social norms on corruption | −0.18 ** | −0.49 *** | 0.57 *** | 0.52 *** |
| Possibilities to engage in corruption | 0.06 | −0.39 *** | −0.14 * | −0.20 * |
| Possibilities to refrain from corruption | −0.23 *** | −0.39 *** | 0.39 *** | 0.34 *** |
| Costs of corruption | −0.24 *** | −0.22 *** | 0.26 *** | 0.26 ** |
| Benefits of corruption | 0.19 ** | −0.34 *** | −0.10 | −0.22 * |
| Social desirability | −0.26 *** | −0.35 *** | 0.10 | 0.29 *** |

Notes: * $p < 0.05$, ** $p < 0.01$, *** $p < 0.001$.

The three models that were tested are displayed in Table 3. The first model confirmed the negative relationship between perceived ethical climate and corruption-proneness, when social desirability was controlled for. The second model indicated that, when all the motives for corruption were included into a single model, personal and social norms on corruption and perceived opportunities to refrain from corruption were the only significant predictors of corruption-proneness. This suggests that perceived opportunities to engage in corruption and the perceived costs and benefits of corruption did not significantly explain corruption-proneness when other motives were controlled for. The full model, including both perceived ethical climate and motives for corruption, revealed that the direct effect of perceived ethical climate on corruption-proneness, indeed, weakened and became statistically non-significant when motives for corruption were included in the model as well. Hence, motives for corruption may, indeed, function as mediators in the relationship between perceived ethical climate and

---

[20]   We also performed a partial correlation analysis, which showed that, in the private sector, perceived ethical climate, motives for corruption, and corruption-proneness were all still significantly correlated when social desirability was included as a covariate. In the public sector, however, perceived ethical climate was no longer significantly related to perceived opportunities to engage in corruption, while social norms on corruption and the perceived benefits of corruption were no longer significantly related to corruption-proneness when social desirability was controlled for.

[21]   Correlation analysis showed that social desirability was also related to all motives for corruption (in both sectors).

corruption-proneness (see Figure 1).[22] To test this further and for each motive separately, mediation analyses were performed via bootstrapping (Zhao et al. 2010; with 1000 resamples derived from the full sample).

**Table 3.** Binary logistic regression model of corruption-proneness (corruption-prone = 1, not corruption-prone = 0; *n* = 523).

| Factor | Model 1 | | Model 2 | | Model 3 | |
|---|---|---|---|---|---|---|
| | B | Wald | B | Wald | B | Wald |
| Sector | −0.028 | 0.014 | −0.148 | 0.238 | 0.203 | 0.432 |
| Social desirability | −0.710 | 36.133 *** | −0.442 | 11.039 ** | −0.448 | 11.242 *** |
| Perceived ethical climate | −0.438 | 14.864 *** | | | 0.152 | 0.888 |
| Personal norms on corruption | | | −0.429 | 9.862 ** | −0.456 | 10.556 ** |
| Social norms on corruption | | | −0.385 | 8.644 ** | −0.448 | 9.294 ** |
| Possibility to engage in corruption | | | 0.114 | 0.952 | 0.105 | 0.791 |
| Possibility to refrain from corruption | | | −0.320 | 5.374 * | −0.337 | 5.841 * |
| Costs of corruption | | | −0.097 | 0.595 | −0.111 | 0.759 |
| Benefits of corruption | | | .238 | 3.351 | 0.238 | 3.317 |
| Overall fit model 1: −2 Log likelihood = 458.715; Cox and Snel $R^2$ = 0.118; Nagelkerke $R^2$ = 0.187. | | | | | | |
| Overall fit model 2: −2 Log likelihood = 393.136; Cox and Snel $R^2$ = 0.222; Nagelkerke $R^2$ = 0.351. | | | | | | |
| Overall fit model 3: −2 Log likelihood = 392.233; Cox and Snel $R^2$ = 0.223; Nagelkerke $R^2$ = 0.353. | | | | | | |

Notes: * $p < 0.05$, ** $p < 0.01$, *** $p < 0.001$.

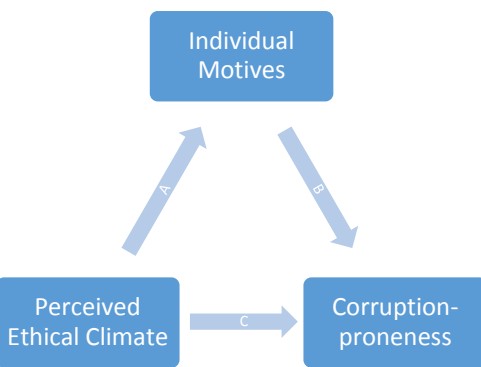

**Figure 1.** Model of individual motives as mediators of the ethical climate–corruption relationship.

Table 4 displays the mean indirect effect of motives for corruption (*a* × *b*) and the 95% confidence intervals (see Figure 1). Table 4 also depicts the *a* and *b* coefficients, as well as the direct effects of perceived ethical climate on corruption-proneness (*c*) and the 95% confidence intervals, to determine whether full or partial mediation was found. The results suggest that personal norms on corruption mediated the relationship between perceived ethical climate and corruption-proneness, as the 95% confidence interval of the indirect effect excluded zero. Table 4 further shows that the 95% confidence interval of the direct effect did include zero, which means that the direct effect of perceived ethical climate on corruption-proneness was no longer statistically significant when personal norms were controlled for. This suggests full cf. indirect-only mediation, referred to by Zhao et al. (2010) as the "gold standard" (p. 198). The same results were found with regard to social norms on corruption: the indirect effect was significant but the direct effect was not. With regard to the other motives for corruption, Table 4 shows that all motives mediated the effect of perceived ethical climate on

---

22   The pattern of results was similar for both men and women.

corruption-proneness, as the 95% confidence intervals of all indirect effects excluded zero. However, the 95% confidence intervals of the direct effects of perceived opportunities to engage in and to refrain from corruption and the costs and benefits of corruption excluded zero as well, which suggests partial mediation. To establish the type of mediation, Zhao et al. (2010) proposed calculating the product of $a \times b \times c$, where if the outcome is positive, this points to complementary mediation, indicating partial mediation. Since all products were positive, the results suggest that perceived opportunities for engaging and refraining from corruption and the perceived costs and benefits of corruption only partially mediated the relationship between perceived ethical climate and corruption-proneness.

**Table 4.** Results of the mediation analyses ($n$ = 523).

| | $a \times b$ | 95% CI of $a \times b$ | $a$ | $b$ | $c$ | 95% CI of $c$ |
|---|---|---|---|---|---|---|
| Personal norms on corruption | −0.226 | [−0.341, −0.141] | 0.351 | −0.644 | −0.207 | [−0.462, 0.047] |
| Social norms on corruption | −0.374 | [−0.534, −0.219] | 0.538 | −0.694 | −0.064 | [−0.335, 0.207] |
| Possibilities to engage in corruption | −0.051 | [−0.111, −0.010] | −0.151 | 0.334 | −0.402 | [−0.627, −0.176] |
| Possibilities to refrain from corruption | −0.190 | [−0.300, −0.104] | 0.315 | −0.601 | −0.264 | [−0.508, −0.019] |
| Costs of corruption | −0.079 | [−0.152, −0.030] | 0.241 | −0.330 | −0.369 | [−0.600, −0.138] |
| Benefits of corruption | −0.051 | [−0.110, −0.014] | −0.122 | 0.413 | −0.408 | [−0.635, −0.181] |

The previous analyses were performed over both groups (while controlling for sector) to enhance statistical power, as well as to provide an overall view on whether, and if so which, individual motives mediated the relationship between perceived ethical climate and employees' corruption-proneness. To examine whether motives functioned as mediators in both the public and private sectors, the same analyses were conducted for the two sectors, separately. The results showed that personal norms on corruption fully mediated the relationship between perceived ethical climate and corruption-proneness, in both the public and private sectors; the mean indirect effects of personal norms were negative and significant, while the direct effects were not (see Table 5). With regard to social norms on corruption, however, the results indicated that social norms functioned as a mediator in the private but not in the public sector. In the private sector, the indirect effect was significant, whilst the direct effect was not, which indicates that social norms fully mediated the effect of perceived ethical climate on corruption in the private sector. Yet, in the public sector, neither the indirect nor the direct effect was statistically significant. Regarding other motives for corruption, the analyses reveal full (cf. indirect only) mediation effects of perceived opportunities to comply, in both sectors, and of perceived costs of corruption, in the public sector. Further, the results confirmed the partial (cf. complementary) mediation effect of perceived opportunities to engage in corruption and the perceived costs and benefits of corruption in the private sector, but not in the public sector.

**Table 5.** Results of the mediation analyses for the public sector ($n$ = 234) and private sector ($n$ = 289).

| | Public Sector | | | | | Private Sector | | | | |
|---|---|---|---|---|---|---|---|---|---|---|
| | $a \times b$ | 95% CI | $a$ | $b$ | $c$ | $a \times b$ | 95% CI | $a$ | $b$ | $c$ |
| Personal norms | −0.173 | −0.317, −0.061 | 0.287 | −0.601 | −0.221 | −0.268 | −0.431, −0.135 | 0.397 | −0.675 | −0.191 |
| Social norms | −0.085 | −0.306, 0.145 | 0.575 | −0.147 | −0.310 | −0.573 | −0.862, −0.358 | 0.513 | −1.118 | 0.150 |
| Possibilities to violate | 0.006 | −0.037, 0.071 | −0.148 | −0.042 | −0.398 * | −0.105 | −0.235, −0.007 | −0.160 | 0.654 | −0.455 ** |

**Table 5.** *Cont.*

|  | Public Sector | | | | | Private Sector | | | | |
|---|---|---|---|---|---|---|---|---|---|---|
|  | $a \times b$ | 95% CI | $a$ | $b$ | $c$ | $a \times b$ | 95% CI | $a$ | $b$ | $c$ |
| Possibilities to comply | −0.152 | −0.352, −0.019 | 0.347 | −0.438 | −0.254 | −0.201 | −0.358, −0.087 | 0.280 | −0.720 | −0.285 |
| Costs of corruption | −0.118 | −0.262, −0.020 | 0.254 | −0.462 | −0.275 | −0.055 | −0.142, −0.008 | 0.221 | −0.250 | −0.429 ** |
| Benefits of corruption | −0.019 | −0.085, 0.010 | −0.068 | 0.286 | −0.373 * | −0.081 | −0.183, −0.014 | −0.168 | 0.481 | −0.437 ** |

Notes: * $p < 0.05$, ** $p < 0.01$.

## 4. Discussion

The purpose of this study was to examine how the ethical climate of organizations, in concert with individual motives for corruption, affects corruption, and bribery, in particular. The findings revealed a relationship between the perceived ethical climate of both public and private organizations and corruption. The more ethical and less egoistic the organizational climate was perceived to be, the less prone employees were to engage in corruption. Our study, therefore, provides the first empirical evidence for a relationship between ethical climate, a general organizational factor, and a specific type of unethical behavior, that is, corruption. Furthermore, in line with another previous study (Gorsira et al. 2016), the results suggest that personal and social norms on corruption were the most important individual-level predictors of employees' proneness to corruption. Extending previous research, we examined whether ethical climate appealed to or affected individual motives for corruption, and whether individual motives, in turn, affected whether or not employees engaged in corruption. As expected, the ethical climate–corruption relationship was fully mediated by personal norms on corruption (in both public and private sector organizations) and social norms on corruption (in private sector organizations). Moreover, the relationship between ethical climate and corruption was partially mediated by the other motives for corruption, the perceived costs and benefits, and perceived opportunities to engage in, and to refrain from, corruption. Hence, the perceived ethical climate seemed to be particularly linked to corruption *through* personal and social norms on corruption. This suggests that public officials and business employees who perceive their organizational climate as more egoistic (i.e., who perceive that self-interest is the dominant consideration within their organization in deciding what constitutes right behavior) feel less morally obliged to refrain from corruption, which in turn increases their proneness to corruption. In addition, private sector employees who perceived their organizational climate as more egoistic believed that corruption was more approved of and more common among their immediate co-workers, which, in turn, appeared to make them more corruption-prone. These results suggest that individual motives for engaging in corruption are the mechanisms through which a general organizational factor, ethical climate, impacts corrupt conduct. Interestingly, normative motives seemed to mediate the relationship between ethical climate and corruption in both the public and private sectors, which implies that normative motives account for the relationship between ethical climate and engagement in both active and passive bribery. This study followed a correlational design, which means that causal inferences are difficult to draw (Andvig et al. 2001; Tavits 2010). Notably, although it is, theoretically, more plausible that general ethical climate affects individual motives for corruption as a specific type of unethical behavior, which in turn influence proneness to corruption, our study does not allow for firm conclusions about causality. Therefore, we cannot rule out that personal and social norms on corruption affect perceptions of an organization's ethical climate, or that a third variable, for instance past corrupt behavior, affects both ethical climate perceptions and individual corruption motives. Longitudinal and experimental studies are necessary to test the causal ordering more thoroughly. Yet, a recent

experimental study provides the first evidence that business culture causally affects employees' unethical behavior (Cohn et al. 2014). Another experimental study found that perceived ethical climate causally affected the corrupt decisions of employees (Gorsira 2018, doctoral dissertation). These experimental studies provide initial experimental evidence that ethical climate precedes corruption, providing further support for our theoretical model. However, more research is required to test whether ethical climate affects the likelihood of corruption via motives for corruption, and whether these, in turn, affect corrupt decisions. Yet, while more experimental research is needed, experiments can only test a limited number of variables at a time, and can create artificial situations that do not represent real-life situations (Sequeira 2012). In contrast, questionnaires enable us to gain insight into multiple key correlates of corruption by surveying large relevant samples, thereby providing insights into respondents' proneness to corruption in the real world. Therefore, it is important to study corruption and its explanatory factors using multiple methods, as different methods have their own strengths and weaknesses and can, thus, provide convergent evidence for the issue at stake (Abbink 2006). Another limitation of the present study was that all data originated from the same method and the same respondents at one point in time. Therefore, testing of the hypotheses may have suffered from common method bias, which usually results in an inflation of observed relationships (Peterson 2002; Podsakoff and Todor 1985). This may have affected our results. We considered the risk of common method bias, and, therefore, guaranteed anonymity and counterbalanced the question ordering, which are design techniques to counteract the influence of common method bias (Podsakoff and Todor 1985; Conway and Lance 2010). Moreover, as social desirability tendencies are another potential source of common method bias, we included a social desirability scale in the questionnaire (Podsakoff and Todor 1985). We have attempted to control for such biases by including a social desirability scale in the relevant regression models. The results indicated that both public and private sector respondents exhibited high social-desirability scores. Moreover, the results showed that social desirability was strongly related to the key factors of interest. Yet, even when we controlled for people's social desirable response tendencies, we found consistent support for our hypotheses. This suggests that the current study may have adequately tapped into the relationships between ethical climate, individual motives for corruption, and proneness to corruption.

We relied on respondents' perceptions to assess their organization's ethical climate. Perceptions may not always reflect the reality that would be observed by outsiders or captured through more objective measures (Martin et al. 2014). Hence, it is unclear how perceptions corresponded to *actual* ethical climates. Although research suggests that employees tend to share their perceptions of the organization's ethical climate, to some extent (Schneider 1975; Wang and Hsieh 2012), it is unclear whether respondents' perceptions of ethical climate were actually shared by others within their organization or department. In the end, however, the individuals' perception is what matters most, as people act upon their perception of a setting more so than upon the actual setting (Wikström 2004).

Our study was conducted in the Netherlands. Future research is needed to examine the extent to which similar results would be found when conducting this study in other cultures and societies. In doing so, researchers could, additionally, examine to what extent and how macro factors affect the individual and organizational factors we studied, and how the three types of factors affect corruption together. Moreover, we focused on only one organizational characteristic, perceived ethical organizational climate, which appeared to be a relevant factor. Future studies could explore the role of other characteristics, both *within* organizations, such as organizational strategy and organizational structure (Huisman 2016), and *outside* the organizations, such as market conditions, like fierce competition, and legal regulations and provisions (Bennet et al. 2013; Luo 2005). Notably, the latter factors may affect both the organization's ethical climate, and the individual motives for corruption of individual employees.

The current study has important practical implications. It suggests a considerable shift in thinking about the causes of corruption and corruption-control initiatives is needed. In their review on corruption research, Andvig et al. (2001, p. 39) concluded that "in recent years, economic

explanations of corruption have been the most cited and probably also the most influential for policy formulations" (see also (Prabowo 2014; Shover and Bryant 1993; Dimant 2013; Svensson 2005)). Present organizational anti-corruption approaches appear to rest heavily on deterrence—detecting and punishing transgressions—and on diminishing opportunities for engaging in corruption—e.g., the 'four-eyes-policy'. The results of the present study do not provide strong support for the assumptions underlying this approach. Economic motives, the perceived costs and benefits of corruption, and perceived opportunities for engaging in corruption did not significantly contribute to explaining proneness to corruption; their influence was outweighed by other individual motives, notably personal and social norms and perceived possibilities for refraining from corruption (see also Gorsira et al. 2016). Hence, in spite of the anti-corruption approach that is now in vogue, our results suggest that combating corruption should not solely be focused on raising costs, diminishing benefits, and reducing opportunities (see also (Hess 2015; Treviño et al. 1999)).

A potentially effective way to motivate people to refrain from corruption is to bolster personal and social norms towards it, since personal and social norms seem to be important predictors of corruption, as well as important routes through which ethical climate affects corruption. Measures focused on norms may be more effective than traditional approaches that focus on threat of detection and severe fines. Organizations can activate personal and social norms through the use of normative messages, for example: "In this organization, people refrain from corruption", addressing the social norm on corruption (De Groot et al. 2013). However, such a message is only likely to lead to stronger social norms to refrain from corruption when people in that organization generally, indeed, refrain from corruption; when they do not refrain from it, such obviously incorrect information might very well backfire and lead to mistrust of the messenger and reinforcing the corrupt norm. In that case, increasing the saliency and strength of personal norms, instead of social norms, might work better, for example: "Do *you* care about honest decision-making? Do not act corruptly" (De Groot et al. 2013; Schultz et al. 2007). Interestingly, one study suggests that implicating the *self* in normative messages is more effective than focusing on the *action* (Bryan et al. 2013). Hence, a normative message might achieve its maximum effect when it is rephrased as follows: "Do *you* care about honest decision-making? Do not be corrupt". Another potentially effective way to encourage employees to hold and to act upon strong personal norms is the use of commitment strategies. Organizations can, for instance, request their employees sign an honor code before, instead of after, they engage in corruption-sensitive tasks, so that normative demands can be made salient at the right place and time (Mazar et al. 2008). Similarly, organizational members can be requested to take a professional oath, preferably in the presence of others (Cohn et al. 2014). An example of such an oath is the bankers oath that was introduced in the Netherlands after the recent financial crises. In this oath, bankers make a promise to execute their function in an ethical manner (Boatright 2013). Commitment strategies are built on the assumption that once people commit themselves to behave ethically, they are motivated to act in line with the promise they made, as they want to (appear to) be consistent (Abrahamse and Steg 2013; Steg 2016). Interventions like these can easily be implemented in both the public and private sectors. Moreover, since people's engagement in both active and passive bribery seems to be affected by normative motives, normative messages and commitment strategies might effectively target both sides of this illegal and unethical act.

Importantly, the present study suggests that corruption should not be seen as an isolated or purely personal issue. When looking at why employees engage in corruption and how to prevent it, organizations should, as well, pay attention to their ethical climate (see also (Hess 2015; Treviño et al. 1999)). The outcomes of the current study indicate that employees' personal and social norms on corruption depend on their organizations' ethical climate. Hence, without policies directed at the organizational level, the aforementioned strategies might be less effective, since an organizational climate that is perceived as egoistic may undermine the strength of employees' personal and social norms on corruption and, thus, encourage corruption.

Organizations can influence their ethical climate, for instance, by paying attention to the ethical issues employees may face in the workplace, by stimulating open discussion about these issues (e.g., by organizing interactive discussions) and by emphasizing which criteria, in the organization's view, should prevail in deciding on these issues. More specifically, organizations may reduce corruption risks by encouraging decision-making based on concern for the wellbeing of others and based on ethical principles, and by simultaneously discouraging an "everyone for him-/herself" atmosphere (Kish-Gephart et al. 2010). Leadership may matter here, as leaders may shape and reinforce employees' perceptions of the organization's ethical climate (Peterson 2002; Hess 2015; Dickson et al. 2001; Mayer et al. 2010; Schminke et al. 2007). To monitor if strategies targeted at the organization's ethical context have the desired effect on employees' perception, organizations can gauge their ethical climate using ethical climate questionnaires (e.g., Victor and Cullen 1988; Arnaud 2010).

The present study suggests that corrupt behavior, similar to other unethical behavior (Wikström 2004; Treviño 1986), is ultimately the result of the interplay between factors at different levels of analysis. However, it is at the organizational and the individual level, and not the country level, that organizations can take the necessary steps to reduce corruption. Particularly, interventions that cultivate an ethical organizational climate, in combination with practical tools that further strengthen personal and social norms on corruption, may contribute to the dwindling of this unethical and illegal behavior. For organizations that have, unfortunately, been confronted with corrupt employees, it seems unwise to exclusively concentrate on the 'bad apples' involved. It is key to not forget the barrel itself, or the public or private organization harboring a corrupt employee.

**Acknowledgments:** The study was funded by PricewaterhouseCoopers Advisory N.V. (PwC The Netherlands).

**Author Contributions:** M.G. and A.D. conceived and designed the experiments; M.G. performed the experiments; M.G. and L.S. analyzed the data; M.G., L.S., A.D. and W.H. wrote the paper.

**Conflicts of Interest:** The authors declare no conflict of interest.

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
