# Peer review of "Corruption in Organizations: Ethical Climate and Individual Motives"

_admsci, doi:10.3390/admsci8010004_

Round 1

Reviewer 1 Report

Below are my comments that, if incorporated, would improve the paper.

1.      H2A: “The more ethical and the less egoistic public and private sector employees perceive their organizational climate to be, the stronger their personal norms and social norms to refrain from corruption…” seems to be a circular statement. The organizational climate itself reflects the social norms. My understanding is that this hypothesis states something along the lines: “The more ethical and the less egoistic public and private sector employees perceive their organizational climate to be, the stronger their personal norms to refrain from corruption and their perception regarding social norms against corruption”. In any case, the authors need to restructure this sentence to clarify what exactly their hypothesis is.

2.      A number of studies have reported gender-differences in attitudes regarding corruption (see for example, Swamy et al., 2001, and more recently, Jha and Sarangi, 2015). It would be interesting to see whether gender matters in the context of this paper. Authors can simply add an indicator for gender in their regression specifications. These studies will provide the motivation and the most recent findings on the association between gender and corruption. This will make the story richer.

3.      Section 4 (materials and methods) can be shortened and should appear before section 2 (results). The reader has no idea what these variables mean until the end of the paper and results make much less sense than it would if the variables are defined before results are presented. In fact, the organization of the paper can greatly be improved.

4.      In the most prominently mentioned explanations for corruption, while incentives, opportunities, and norms are noted, two very important factors – culture (see for example, Jha and Panda, 2017) and political institutions are not mentioned (see for example, Lederman et. al., 2005) – are missing. It would be helpful for the readers if these studies are mentioned. The culture is important even in the context of the organization factors. If the culture (in a country or within an organization) is accepting of corruption, then an individual is likely to involve in bribery because there is no social stigma or social cost. The author should include a brief discussion of the importance of culture on the prevalence of corruption. For instance, Jha and Panda (2017) find that individualist societies are less corrupt than collectivist societies for a number of reasons including the consideration of self-interest versus the interest of the society. Since the present paper discusses this line of reasoning, it seems that authors may find further motivation for their study from this paper. Given that the paper is about the ethical notion and personal and social norms, the complete absence of the mention of the influence of culture on corruption is surprising.

Minor comments

1.      At several places on page 2, paper is cited as (XXX, 2016).

2.      Line 219: “than” should be replaced with “that”.

3.      Line 314: citation needs to be fixed.

Author Response

We uploaded our reply to the Review Report as a PDF file

Reviewer 2 Report

The authors need to insert references in several places (e.g., xxx, 2016). Hypothesis 2a is tautological and I recommend dropping it. Materials and Methods should appear before Results and Conclusion.

Author Response

We uploaded our reply to the review report as a PDF file. 
